# Elevated Mortality Risk from CRKp Associated with Comorbidities: Systematic Review and Meta-Analysis

**DOI:** 10.3390/antibiotics11070874

**Published:** 2022-06-29

**Authors:** Lucas Candido Gonçalves Barbosa, José Arthur Silva e Sousa, Graziela Picciola Bordoni, Gabriel de Oliveira Barbosa, Lilian Carla Carneiro

**Affiliations:** 1Institute of Tropical Pathology and Public Health, Federal University of Goiás (UFG), 235 Street, Neighborhood: Leste Universitário, Goiânia 74605-050, Brazil; lucascandidogomesalves46@outlook.com (L.C.G.B.); grazielapicciola@discente.ufg.br (G.P.B.); 2National Institute of Courses, R Six Street, 85 Number, Neighborhood: Oeste, Goiânia 74125-080, Brazil; josearthur231@gmail.com; 3Biology Department, FAVENI College, Ângelo Altoé Street, 888 Number, Neighborhood: Venda Nova do Imigrante, Goiânia 29375-000, Brazil; gabrieloliveira1296@live.com

**Keywords:** mortality, health public, *Klebsiella pneumoniae*, carbapenems

## Abstract

Carbapenem-resistant *Klebsiella pneumoniae* has become a public health problem with therapeutic limitations and high mortality associated with comorbidities. Methods: This is a systematic review and meta-analysis with a search in PubMed, SciELO, and Lilacs. Ten articles were selected, considering cohort, case-control, and cross-sectional studies. Tests for proportions and relative risk of mortality were performed, considering a 5% threshold for significance. Statistical analyses were performe dusing Rstudio^®^ software, version 4.0.2 of Ross Ihaka and Robert Genleman in Auckland, New Zealand. Results: *Klebsiella pneumoniae*, associated with chronic kidney disease, was responsible for 26%/258 deaths, chronic lung disease 28%/169, diabetes 31%/185, liver disease 15%/262, and heart disease 51%/262 deaths. Carbapenem-resistant *Klebsiella pneumoniae* associated with chronic kidney disease was responsible for 49%/83 deaths, with diabetes 29%/73, and with liver disease 33%/73 deaths. The risk of death from carbapenem-resistant *Klebsiella pneumoniae* was twice as high as the number of deaths associated with carbapenem-sensitive *Klebsiella pneumoniae*, RR = 2.07 (*p* < 0.00001). Conclusions: The present study showed an increase in mortality from carbapenem-resistant *Klebsiella pneumoniae* when associated with comorbidities.

## 1. Introduction

Antibiotic resistance is a major health problem, especially when it comes to the control of infections caused by microorganisms that were sensitive to antimicrobials. Many microorganisms, especially bacteria, have become resistant to almost all clinically important antimicrobials. The biggest problem is that the pharmaceutical industry does not have enough new drugs to complement the growth of resistance of these microorganisms, such as *K. pneumoniae*. The growing number of cases of antibiotic-resistant *Klebsiella pneumoniae* infections stands out as a worldwide health problem, presenting as a clinically important bacterium causing pneumonia in hospital environments [1].

The resistance of *K. pneumoniae* is a consequence of the presence of extended-spectrum β-lactamases (ESBLs), described as enzymes that, through plasmid mediation, confer resistance to penicillins and cephalosporins, including combinations of sulbactam, clavulanic acid, and monobactams. ESBLs are commonly detected in strains of Klebsiella pneumoniae, qualified as an opportunistic pathogen, the etiologic agent of serious infections in hospitalized patients, with emphasis on patients with compromised immunity to serious diseases. *K. pneumoniae* capable of producing β-lactamase was first detected in Germany in 1983, and there has been a steady increase worldwide in subsequent decades [1,2].

Nosocomial infections are usually caused by *K. pneumoniae*, *Acinetobacter baumanie,* and other gram-negative microorganisms. These bacteria used to cause a variety of serious health problems all over the world. Carbapenem-resistant Enterobacteriaceae (CRE) infections, in particular, have become a constant challenge for therapeutics [2,3,4].

Among the CRE, *Citrobacter freundii*, *Escherichia coli,* and carbapenem-resistant *K. pneumoniae* (CRKp) [5] stand out. According to the Clinical and Laboratory Standards Institute, CRKp are all *K. pneumoniae* isolates resistant to any carbapenems: meropenem, imipenem, or ertapenem, with this, CRKp emerges as a multidrug-resistant microorganism resistant to most antibiotics currently availablewith a great threat to human health [6,7].

CRKp infections are treated with restricted therapeutic options, such as aminoglycosides, tigecycline, colistin, and cetftazidime associated with avibactam. Resistance to carbapenems is a consequence of the presence of genes encoding carbapenemases, among the CRKp isolates, the *bla*KPC-2 and *bla*KPC-3 genes are the most common [8,9,10,11].

CRKp is responsible for a high mortality rate, aggravated by the significant increase in the worldwide prevalence rate [11]. The rapidly increasing morbidity rate of the pathogen is a consequence of clonal and plasmid-interceded dissemination of strains not sensitive to carbapenems [5,9,10,12,13]. Considering the high therapeutic and clinical costs associated with CRKp infections, in view of the mechanisms of resistance and unavailability of effective treatment, the World Health Organization (WHO) recognized CRKp as a critical priority level 1 bacterium for the development of new antibiotics [14,15].

Several infectious diseases are associated with adjacent diseases, increasing the risk of infection and complications. Epidemiological information shows that the increased incidence of nosocomial infections and some community-acquired infections is related to those patients with preexisting comorbidities, such as diabetes, immunosuppressant, lung diseases, and nephropathies [1,14,15].

Most studies related to CRKp focus on presenting epidemiological characteristics, mainly in relation to geographic distribution, and phenotypic and molecular characterization of CRKp strains. However, there is a lack of studies that seek to evidence the proportion of mortality associated with CRKp and comorbidities. Therefore, the aim of this study was to determine the mortality rate of patients infected with CRKp associated with comorbidities and to assess the relative risk of mortality in patients with CRKp and with carbapenem-sensitive *Klebsiella pneumoniae* (CSKp).

## 2. Results

### 2.1. Proportion of Deaths from K. pneumoniae Associated with Comorbidities

The analysis of bias through the Beg test showed a risk of bias without significance, with *p* = 0.22. The systematic review resulted in the tabulation of the following data: for the analysis of the mortality subgroup of patients with kidney disease associated with *K. pneumoniae*, the combined sample number was 258 patients, for the mortality subgroup of patients with pulmonary diseases associated with a *K. pneumoniae*: *n* = 169, for the subgroup mortality of patients with diabetes associated with *K. pneumoniae*: *n* = 185, for mortality of patients with hepatic diseases associated with *K. pneumoniae* infection: *n* = 262, and for mortality of patients with heart disease associated with *K. pneumoniae* infection: *n* = 262.

The results showed a large proportion of deaths caused by *K. pneumoniae* when associated with comorbidities. The mortality rate of patients with chronic kidney disease infected with *K. pneumoniae* was 26%, prop = 0.26 (CI95% = 0.10–0.53; I² = 88%). Chronic lung disease associated with *K. pneumoniae* infection had a proportion of deaths equal to 28%, prop = 0.28 (0.15–0.45; I² = 69%). For diabetes associated with *K. pneumoniae*, the proportion of deaths was 31%, prop = 0.31 (0.25–0.38; I² = 39%). Liver disease associated with *K. pneumoniae* infection had a 15% proportion of deaths, prop = 0.15 (0.11–0.19; I² = 33%), and heart disease associated with *K. pneumoniae* infection had a 51% proportion of deaths, prop = 0.51 (0.35–0.68; 87%) (Figure 1).

### 2.2. Proportion of Deaths by CRKp

The systematic review of mortality of patients with comorbidities associated with CRKP infections presented the following data: for the mortality of patients with kidney disease associated with CRKP infection, the number of patients after combining the data was 83, for the proportion of deaths of patients with diabetes associated with CRKP infection: *n* = 73, for mortality of patients with liver diseases: *n* = 73.

The results showed a high proportion of mortality in patients with chronic kidney disease when associated with CRKP, prop = 0.49 (0.39–0.60; I² = 0%), that is, approximately 50% of patients had a fatal outcome. For patients with diabetes, the proportion of deaths associated with CRKP was 29%, prop = 0.29 (0.20–0.40). The proportion of deaths of patients with CRKP-associated liver disease was 23%, prop = 0.23 (0.05–0.61) (Figure 2).

### 2.3. Risk of Death from CRKp Compared to CSKp

For the assessment of relative risk, the systematic review resulted in the following data: For the CRKP group, 387 patients were considered, of these, 208 died. The CSKP group consisted of 887 patients, of whom 183 died. The risk of death related to CRKP was higher when compared to deaths from CSKP, with RR = 2.07 (1.39–3.09; I² = 83%); the results obtained indicate a risk of death from CRKP twice as high as CSKP (Figure 3).

## 3. Discussion

The presence of resistant bacteria stands out as a frequent problem in hospital environments. The emergence of multidrug-resistant species is related to the increase in resistance of members contained in the Enterobacteriaceae family. The main form of resistance among gram-negative bacteria is the ability to produce beta-lactamases which are responsible for bacterial resistance to beta-lactams. Beta-lactamases are enzymes responsible for the degradation of the beta-lactam ring, thus inactivating antimicrobials and preventing activity against the enzymes responsible for the synthesis of the bacterial cell wall. Among the bacteria producing beta-lactamases, the genus *Klebsiella* spp, *Escherichia* spp, *Enterobacter* spp, *Salmonella* spp, *Serratia* spp, *Citrobacter* spp, *Proteus* spp, and *Morganella* stand out. The genes responsible for encoding the most prevalent carbapenemases are from the *bla*KPC, *bla*VIM, *bla*IMP, *bla*Oxa, and *bla*Ndm groups [16].

*K. pneumoniae* is responsible for causing one-third of gram-negative infections such as urinary tract infections, pneumonia, cystitis, surgical wound infections, endocarditis, and septicemia. *K. pneumoniae* can also cause necrotizing pneumonia and pyogenic liver abscesses. This microorganism has been associated with high death rates, high treatment costs, and prolonged hospital stays. *K. pneumoniae* is a microorganism present in all environments; it is one of the opportunistic pathogens with high relevance, standing out for causing several infections in humans such as UTIs, surgical infections, and respiratory tract infections causing pneumonia. The development of resistance in *K. pneumoniae* isolates is a consequence of the production of extended-spectrum β-lactamases (ESBLs). *K. pneumoniae* strains that produce ESBL are found all over the world, causing numerous outbreaks of infections. Bloodstream infections caused by *K. pneumoniae* can occur because of ventilator-acquired pneumonia or through social contact, through the urinary system, intra-abdominal diseases, and central venous line infections [16,17,18,19].

CRKP stands out as a major public health concern, as infections caused by these bacteria are highly associated with mortality [16,17]. Mortality in hospital settings can be explained by the fact that patients who become infected with CRKP are, in general, acutely ill and chronically ill (with comorbidities). Comorbidities can lead patients to longer exposure time in hospitals, hospitalizations, and prolonged use of antibiotics, which are relevant risk factors for CRKP infection [18]. The present meta-analysis confirms the strong association between the increase in mortality of patients with confirmed CRKP, the proportion of deaths, and the increase in risk.

Patients with three or more comorbidities demonstrated an independent risk factor for CRKP infections and were associated with a higher rate of ICU admission, worse clinical outcomes, and mortality [16,19]. Patients with comorbidities are usually in a precarious physical condition, in addition to having an immune system with compromised function in certain situations and being more prone to infections with high severity because they do not have sufficient defenses against resistant infections.

The multicenter study by Bahlis and collaborators [19] carried out in China, used the Charlson Comorbidity Index (ICC), which is applied to classify the severity based on comorbidities leading to a higher risk of death to the patient. In this study, 831 patients had CRKP infections, and the mean ICC found was 5, demonstrating a high risk of death in patients with CRKP [19,20]. The results of existing research point to the assertiveness of the present systematic review, however, the meta-analysis in question presents mortality results from CRKP and CSKP, contributing with relevant information about the current situation experienced in hospital environments, with an emphasis on ICUs.

In the study by Wang and collaborators [11], 96 patients were evaluated, 48 of whom were infected with CRKp and the other 48 with CSKp. Among patients infected with CRKp, 23 (47.9%) died, while 2 (4.2%) patients in the CSKp infected group died, indicating assertiveness in the results obtained in the present meta-analysis where the increase in the risk of death by CRKp was higher than the risk of death by CSKp. Wang and collaborators [11] also presented that, among the factors associated with death from *K. pneumoniae* infection in both groups, the comorbidities found were cardiovascular disease—20 patients (80%); renal dysfunction—17 patients (68%); neurological disease and diabetes mellitus—12 patients each (48% each); Tumor—11 patients (44%); hepatobiliary disease—6 patients (24%); pulmonary disease—3 patients (12%); and autoimmune disease—2 patients (8%). Although the results by Wang et al. [11] demonstrate similarity to the results of the present review, the reliability of the new results is presented with greater weight, considering the larger sample number after combining the selected data. The data show the correlation between the higher risks of mortality in patients who have some comorbidity in CRKp infection compared to those who do not have comorbidities.

It is believed that the colonization of K. pneumoniae in a community or even in the hospital environment may be related to the location, although there are several other factors. In the study by Ling et al. [21], it was found that the Chinese had a colonization rate of 66% compared to Malays (14.3%), Indians (7.9%), and other nationalities (11.8%). The reason why colonization differs between these populations is not clear, but it may be linked to environmental factors that provide greater exposure to the pathogen.

The present research highlights important data on the increased risk of death, and the outcome of the association between comorbidities and CRKP infection. In future perspectives, the care of the health professional handling equipment and adequate sterilization of the materials used is highlighted to reduce the risk of hospital infection. Despite the importance of the results, the lack of information on the exposure time or site of infection stands out as a limitation. The lack of information on morbidity or causes of diseases is also considered a limitation.

## 4. Materials and Methods

### 4.1. Kind of Study

The present research is characterized as a systematic review, associated with meta-analysis, with a search in the following databases: SciELO, Lilacs, the website of the National Center for Biotechnology Information (NCBI), PubMed, and Medline. The search for references was performed using keywords found on the Health Sciences Descriptors website (DeCS/MeSH) considering Mortality, Health Public, *Klebsiella pneumoniae*, Carbapenems. The descriptors were associated with the Boolean operators AND, OR and NOT.

In PubMed and MEDLINE, 255 references were found, in Scielo 35, and Lilacs 13. Duplicate articles and articles that did not meet the inclusion criteria were discarded. After reviewing 74 articles, 26 articles were excluded because they did not fit the proposed theme and 7 were not retrieved. After reviewing the articles, 34 were selected for this research. Of these 34 articles, 11 were selected for meta-analysis, according to the Preferred Reporting Items for Systematic Reviews and Meta-Analyses set of items [22] (Figure 4). The evaluation process of the data extracted for meta-analysis was carried out by two statistical reviewers, there were no objections regarding the selected data.

### 4.2. Inclusion and Exclusion Criteria

For the descriptive aspect, studies published in the last four years (2018 to 2022) were considered, however, for the systematic review there was no time limitation. Inclusion criteria were cohort-type studies, cross-sectional studies, and studies that addressed mortality from CRKP or CSKP. Taking into account comorbidities, data were included when participants had diabetes, lung disease, heart disease, liver disease, and kidney disease. There was no language limitation; however, the selected studies should present reliable methodologies. Exclusion criteria were research with dubious methodologies, unfinished research, master’s dissertations, and doctoral theses.

### 4.3. Data Extraction and Treatment

Data from cohort, case-control, and cross-sectional surveys were considered and extracted. Cohort data were used to analyze the incidence or risk of death from CRKP and CSKP; on the other hand, data from cross-sectional surveys were used to assess the proportion of deaths of patients with K. pneumoniae and CRKP associated with comorbidities (Table 1).

### 4.4. Applied Statistics

To estimate the proportion of mortality, the proportion test was performed, considering the generalized linear mixed model (GLMM), to determine the proportion of mortality in patients with CRKP associated with comorbidities. The relative risk of mortality was also evaluated, comparing patients with CRKP with patients with CSKP. The Higgins and Thompson test (I²) was applied to determine assertiveness heterogeneity in the choice of effects, considering the randomized effect for the analysis [34]. A threshold of 5% for significance was considered. Statistical analyses were performed using RStudio^®^ 4.0.2 software.

### 4.5. Risk of Bias

The risk of bias was estimated using the nonparametric test of beg, considering the significance limit of 5%. A qualitative analysis of the studies selected for meta-analysis was also performed according to information extracted from the methodologies. All selected studies were from participants in ICUs. The studies selected to determine the proportion were qualified as observational, thus, without interference from the researchers.

## 5. Conclusions

*K. pneumoniae* is associated with a high mortality rate, standing out as a global health problem. Hospitalized patients with comorbidities almost always present impairment or limitations of the immune system, favoring opportunistic microorganisms. The association between comorbidities and CRKp increased the proportion of deaths and increased the risk of death. This data indicates the need to prioritize the problem addressed.

## Figures and Tables

**Figure 1 antibiotics-11-00874-f001:**
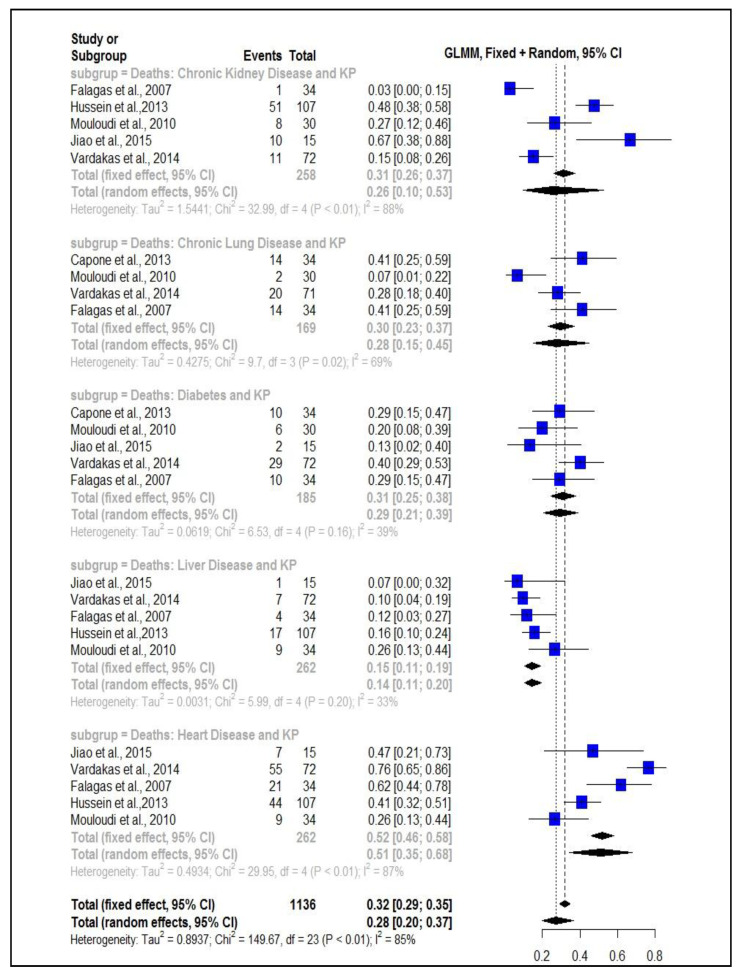
Test for proportion determination with a mixed generalized linear model (GLMM): Mortality ratio of patients with comorbidities associated with *K. pneumoniae*.

**Figure 2 antibiotics-11-00874-f002:**
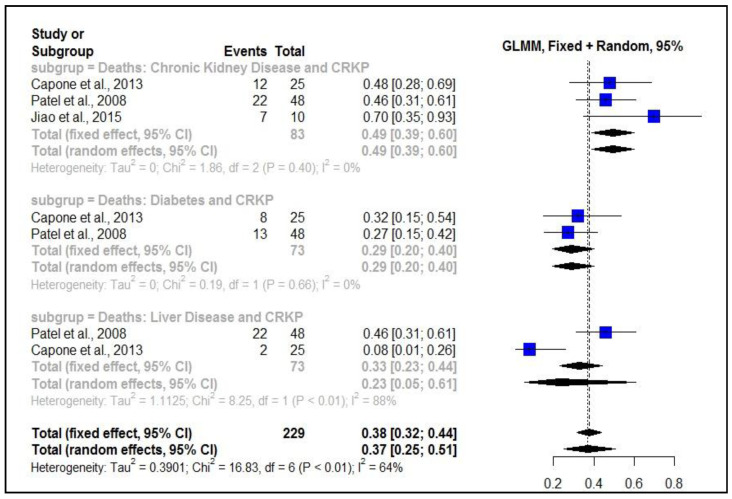
Test for proportion determination with a mixed generalized linear model (GLMM): Proportion of mortality of patients with comorbidities associated with CRKP.

**Figure 3 antibiotics-11-00874-f003:**
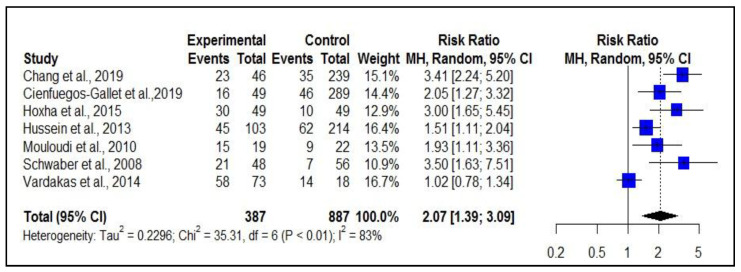
Risk of death of patients with CRKP when compared to patients with CSKP.

**Figure 4 antibiotics-11-00874-f004:**
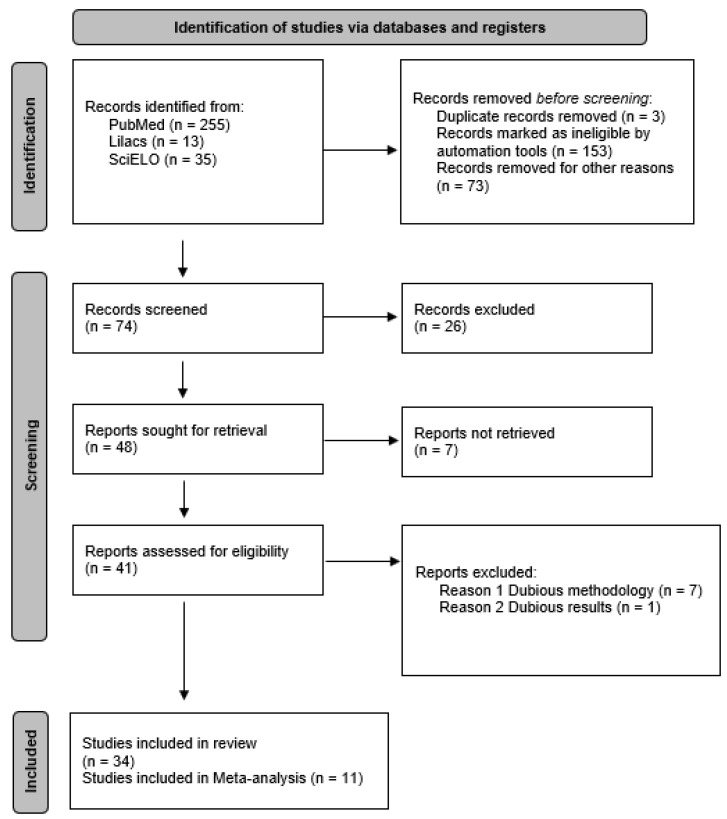
Flowchart for the selection process of articles used in the Meta-analysis.

**Table 1 antibiotics-11-00874-t001:** Research selected for meta-analysis.

Studies Selected for Meta-Analysis			
Author	Year	Countries	Type of Study
Falagas et al. [23]	2007	Greece	Case-control study in two hospitals
Hussein et al. [24]	2013	Israel	Retrospective case-control study
Mouloudi et al. [25]	2010	Greece	Case-control study
Jiao et al. [26]	2015	China	Retrospective case-control study
Vardakas et al. [27]	2014	Greece	Retrospective cohort
Capone et al. [28]	2012	Italy	Retrospectiveobservational study
Patel et al. [29]	2008	USA	Case-control study
Chang et al. [30]	2019	China	Retrospectivecohortstudy
Cienfuegos-Gallet et al. [31]	2019	Colombia	Case-control and a cohort study
Hoxha et al. [32]	2015	Italy	Cohort study
Schwaber et al. [33]	2008	Israel	Case-control and cohort study

## Data Availability

The data are contained in the article.

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
