# Peer review of "Elevated Mortality Risk from CRKp Associated with Comorbidities: Systematic Review and Meta-Analysis"

_antibiotics, 2022, doi:10.3390/antibiotics11070874_

Round 1
Reviewer 1 Report
The meta-analysis work is very interesting and highlights a serious and widespread problem throughout the world.
But I did not understand why the authors inserted the results and the discussion before the materials and methods ... I recommend following the classic drafting of scientific works.
For the rest I report an error I think of typing at line 47 (It are bacteria considered serious ... I think it is: THEY are bacteria considered serious ...) -
Author Response
Comments and suggestions for authors
The meta-analysis work is very interesting and highlights a serious and widespread problem across the world.
But I didn't understand why the authors inserted the results and the discussion before the materials and methods... I recommend following the classic writing of scientific papers.
This article is formatted according to the layout of the journal.
Otherwise I report an error I think about typing on line 47 (These are bacteria considered serious... I think it is: THEY ARE bacteria considered serious...) -
Was modified and highlighted in yellow.

Reviewer 2 Report
In this study, authors analysed mortality risk from CRKp associated with comorbidities. Study is well written, however, some aspects should be further addressed:
It isn't entirely clear why were 26 records excluded after screening.
How was risk of bias of included studies estimated? Detailed rationale needed.
All of the figures are presented with low quality and could be reuploaded with better resolution.
Discussion is missing some important aspects – limitations of this study, as well as clinical implications and future perspectives that could be derived from the presented results.
Author Response
Reviewer 2
Comments and suggestions for authors
In this study, the authors analyzed the risk of mortality from CRKp associated with comorbidities. The study is well written, however, some aspects must be addressed:
It is not entirely clear why 26 records were excluded after screening.
After reviewing 74 articles, 26 articles were excluded because they did not fit the proposed theme and seven were not retrieved.
How was the risk of bias of the included studies estimated? Detailed reasoning required.
The analysis of bias through the begg test showed a risk of bias without significance, with p = 0.22. The risk of bias was estimated using the nonparametric test of beg, considering the significance limit of 5%. A qualitative analysis of the studies selected for meta-analysis was also performed according to information extracted from the methodologies. All selected studies were from participants in ICUs. The studies selected to determine the proportion were qualified as observational, thus, without interference from the researchers.
All figures are presented with low quality and can be resent with better resolution.
Figures have been replaced.
The discussion is missing some important aspects – limitations of this study, as well as clinical implications and future perspectives that can be derived from the results presented.
Added information, highlighted in yellow in the thread.
The present research highlights important data on the increased risk of death, outcome of the association of comorbidities and CRKP infection. With that, as future perspectives, the care of the health professional is highlighted when handling the equipment and adequate sterilization of the materials used, reducing the risk of hospital infection. Despite the importance of the results, the lack of information on the exposure time or site of infection stand out as limitations. The lack of information on morbidity or causes of diseases is also considered limitations.
Reviewer 3 Report
The article “Elevated Mortality Risk from CRKp Associated with Comorbidities: Systematic Review and Meta-Analysis” by Barbosa et al., is interesting. However as per my observation below are a few points, which are missing and need to be clarified.
1. Figure1. is not made clear and there is no heading for various components such as statistics etc.
2. Figure 4. Flowchart for the selection process of articles used shows studies included in review =34. In contrast, the calculation shows thirty-three, and the studies included in the meta-analysis is 11, whether this is among the 33 or excluding 33.
3. The author mentioned the study included the last four years of the published article while not saying the selection of papers from the which-to-which year because readers do not know when the study started when the article was written, and when it will be published.
4. The author does not distinguish whether the patient in the included study were having comorbidities prior to Klebsiella pneumoniae infections or it occurred following hospitalizations of the patients.
5. The authors also did not include in their inclusion criteria, If there are any specific criteria for comorbidities they included in their study or which comorbid patient they preferred.
6. The author didn’t show the length of carbapenem treatment and other drugs in combination.
7. Line 47, “It are bacteria considered serious health problems worldwide” is confusing.
8. Line 203 mentioned “totaling” correct it.
Author Response
Reviewer 3
Comments and suggestions for authors
The article “Elevated Mortality Risk from CRKp Associated with Comorbidities: Systematic Review and Meta-Analysis ” by Barbosa et al., is interesting. However, as per my observation below, there are some points that are missing and need to be clarified.
- Figure1. is unclear and there is no title for various components such as stats etc. Test for proportion determination with mixed generalized linear model:
Was Modified.
- Figure 4. Flowchart of the selection process of the articles used shows the studies included in the review =34. In contrast, the calculation shows thirty-three, and the studies included in the meta-analysis are 11, either among the 33 or excluding 33.
Was Modified.
- The author mentioned that the study included the last four years of the published article, but did not say the selection of articles from which year to which year because readers do not know when the study began, when the article was written, and when it will be published.
For the descriptive aspect, studies published in the last four years (2018 to 2022) were considered. highlighted in yellow.
- The author does not distinguish whether the patient in the included study had comorbidities prior to Klebsiella pneumoniae infection or if it occurred after the patients were hospitalized.
Highlighted in yellow in discussion
- The authors also did not include in their inclusion criteria whether there are any specific criteria for comorbidities that they included in their study or which patient with comorbidity they preferred.
Taking into account comorbidities, data were included when participants had: diabetes, lung disease, heart disease, liver disease and kidney disease. Highlighted in yellow.
- The author did not show the duration of treatment with carbapenem and other drugs in combination. The included studies did not provide data on treatment time.
This was not the purpose of the present meta-analysis.
- Line 47, “Are bacteria considered serious health problems worldwide” is confusing.
Was Modified, highlighted in yellow.
- Line 203 mentioned “totaling” correct.
Was Modified, highlighted in yellow.
Reviewer 4 Report
The aim of this manuscript is to demonstrate the mortality rate of Carbapenem-resistant Klebsiella Pneumonia (CRKP) as compared to Carbapenem-sensitive Klebsiella Pneumonia (CSKP) in patients with different comorbidity by Meta-analysis and found that patient with CRKP had higher mortality rates associate with comorbidities.
1. The mortality rate in patient without comorbidities should be analysis and compared between two groups
2. More details should be included in different comorbidities: Stage of CKD, more specific diagnose of chronic lung disease, chronic liver disease, and heart diseases.
3. Following with the above comments, the mortality rate and relative risk for patient with CRKP and CSKP in more specific diseases should be compared.
4. A list of antibiotics for CRKP should be provided.
Author Response
Reviewer 4
Comments and suggestions for authors
The objective of this manuscript is to demonstrate the mortality rate of Carbapenem-resistant Klebsiella Pneumonia (CRKP) compared to Carbapenem-sensitive Klebsiella Pneumonia (CSKP) in patients with different comorbidities by Meta-analysis and to verify that the patient with CRKP had higher rates of mortality associated with comorbidities.
The objetive of this research is to highlight the proportion of deaths when associated with CRKP and CSKP infection.
- The mortality rate in patients without comorbidities should be analyzed and compared between two groups
The searches selected were not of the case/control type, the odds ratio was not estimated, the selected surveys were of the observational type, with the purpose of verifying proportion of deaths in isolation.
- More details should be included on the different comorbidities: CKD stage, more specific diagnosis of chronic lung disease, chronic liver disease and heart disease.
Limitations was included in the discussion.
- Following on from the comments above, one should compare the mortality rate and the relative risk for patients with CRKP and CSKP in more specific diseases.
For assessment of the risk of death, surveys of the cohort type were included, this analysis is not intended to associate with commodities, but to compare risk of death between CSKP and CRKP.
- A list of antibiotics for CRKP must be provided.
The treatment is very limited, there are not many antibiotics, however, in the introduction, the treatment used is highlighted in yellow.
Round 2
Reviewer 2 Report
The authors have answered the questions and improved the paper, although they could have been more responsive, with presentation of exact lines in text where the changes have been made.
I have no further questions.
Author Response
- The authors have answered the questions and improved the paper, although they could have been more responsive, with presentation of exact lines in text where the changes have been made. I have no further questions.
Author´s answer: The evaluation and analysis of bias was added in results and methodology.
- The author made significant and satisfactory changes as required. However, the line "They are bacteria that cause serious health 50 problems all over the world" (49-50) on the page number 2, needs to be replaced with
Author´s answer: "These bacteria used to cause a variety of serious health problems all over the world". Changed as requested, highlighted in yellow.
- The author response all the comments with some modification.
Author´s answer: Modifications have been made.

Reviewer 3 Report
The author made significant and satisfactory changes as required.
However, the line "They are bacteria that cause serious health 50 problems all over the world" (49-50) on the page number 2, needs to be replaced with
"These bacteria used to cause a variety of serious health problems all over the world".
Author Response
- The author made significant and satisfactory changes as required. However, the line "They are bacteria that cause serious health 50 problems all over the world" (49-50) on the page number 2, needs to be replaced with
Author´s answer: "These bacteria used to cause a variety of serious health problems all over the world". Changed as requested, highlighted in yellow.

Reviewer 4 Report
The author response all the comments with some modification.
Author Response
- The author response all the comments with some modification.
Author´s answer: Modifications have been made.
